# Application of Blockchain Technology in Dynamic Resource Management of Next Generation Networks

**Michael Xevgenis** [1,*] , **Dimitrios G. Kogias** [2] , **Panagiotis Karkazis** [3] , **Helen C. Leligou** [1] and **Charalampos Patrikakis** [2]

1  Department of Industrial Design and Production Engineering, University of West Attica,
   122 43 Attica, Greece; e.leligkou@uniwa.gr
2  Department of Electrical and Electronics Engineering, University of West Attica, 122 43 Attica, Greece;
   dimikog@uniwa.gr (D.G.K.); bpatr@uniwa.gr (C.P.)
3  Department of Information and Computer Engineering, University of West Attica, 122 43 Attica, Greece;
   p.karkazis@uniwa.gr
*  Correspondence: mxevgenis@uniwa.gr

**Abstract:** With the advent of Software Defined Networking (SDN) and Network Function Virtualization (NFV) technologies, the networking infrastructures are becoming increasingly agile in their attempts to offer the quality of services needed by the users, maximizing the efficiency of infrastructure utilization. This in essence mandates the statistical multiplexing of demands across the infrastructures of different Network Providers (NPs), which would allow them to cope with the increasing demand, upgrading their infrastructures at a slower pace. However, to enjoy the benefits of statistical multiplexing, a trusted authority to govern it would be required. At the same time, blockchain technology aspires to offer a solid advantage in such untrusted environments, enabling the development of decentralized solutions that ensure the integrity and immutability of the information stored in the digital ledger. To this end, in this paper, we propose a blockchain-based solution that allows NPs to trade their (processing and networking) resources. We implemented the solution in a test-bed deployed on the cloud and we present the gathered performance results, showing that a blockchain-based solution is feasible and appropriate. We also discuss further improvements and challenges.

**Keywords:** blockchain; distributed ledger technologies; resource management; software defined networks; next generation networks; 5G

## 1. Introduction

Telecommunication networks' infrastructures continuously evolve to optimize the user experience and, at the same time, maximizing the network and service providers' revenues and profits. Major evolution paradigms in this perspective include 5G/6G technologies which aim at significantly improving the experience of the user by improving network bandwidth and, also, by enhancing the resource management mechanisms. In this direction, the adoption of network virtualization technologies such as Software Defined Networking (SDN) and Network Function Virtualization (NFV) has multiple implications on resource management: each Network Service (NS) offered to the user can be split in Virtual Network Functions (VNFs) which are deployed in dynamically-decided virtual infrastructures, enabling resource utilization balancing and management with direct impact on the observed quality of service and resource utilization. The upcoming Tactile Internet, which is based on real time decisions based on Artificial Intelligence (AI) is one among the demanding applications which nowadays see wide expansion. In the context of the Next Generation Internet (NGI), it is crucial

to maximize efficiency in the support of applications with diverse requirements by enabling fully dynamic sharing of resources (both computational and networking) not only inside the infrastructure of the NP but, also, among NPs' infrastructures.

These needs motivated the research and development of various Management and Orchestration (MANO) platforms such as the OSM MANO [1], ONAP [2], SONATA [3]. These opensource platforms implement new management techniques like network slicing, network service lifecycle management and, at the same time, implement the NFV architecture providing a flexible, scalable and robust operation framework. A network slice is an isolated amount of network resources customized to meet specific service requirements. As a result, various Network Services (NS) can operate in the same infrastructure and share the same network capabilities.

Although this can yield interesting results within an administrative area, to achieve the full range of the benefits of this capability, the different parts of the network infrastructure, which are administered by different actors, should be able to dynamically agree on resource usage, sharing and management. In other words, it would be highly beneficial for all networking resource providers to be able to dynamically negotiate computational and network resources in different Points of Presence (PoP) to guarantee Service Level Agreements (SLAs) worldwide. The contracting providers are interested in lowering their expenses (both CAPEX and OPEX) while respecting their contracts with their customers. SLA monitoring software today is deployed at the edges of different administrative domains, in order to ensure the contracted SLAs are respected. With the adoption of the SDN/NFV technologies, the potential to negotiate more fine-grained SLAs emerges, not only among infrastructure and service providers, but also in network service level. This means that service providers can rearrange the SLA with a specific infrastructure provider based on the SLAs of the current active Network Server in the specific infrastructure. However, this is difficult to achieve in a competitive environment at the absence of a trusted 3rd party.

In parallel, with the advances in networking architectures and management solutions, advances take place, also, in distributed trust systems which come under the umbrella of blockchain technologies or Distributed Ledger Technologies (DLT). Blockchain/DLT is one of the most hyped technologies, as it introduces trust in untrusted environments. DLTs (a sub-category of which is blockchain) are gaining in popularity recently, and have two basic forms: either they follow the structure of a Direct Acyclic Graph (DAG) or they place encrypted transactions in blocks that are uniquely connected in a chain to form a Blockchain.

The main characteristic of this technology is that the, so-called, "transactions" between different entities are registered in a digital ledger, a replica of which is kept in all participating nodes. This technology guarantees: (a) the integrity of the information that is kept in the ledger, (b) the immutability of this information, and (c) the sovereignty, which means that the entity that inserted the information is known. Instead of keeping all the "transactions" in a single central point, the ledger is kept in multiple nodes offered by the participating entities. The various blockchain and DLT platforms and solutions that have been proposed in the literature (see for example [4–8]) differ with respect to multiple relevant characteristics, the most important of which are: (a) the public vs. private operation regarding the access to the network, (b) the consensus protocol adopted and (c) the adoption (or not) and the type of information encryption. Appropriately, selecting the different parameters and solutions suitable for different sectors and applications can be defined. For example, solutions for the logistics sectors, for the supply chain and for the FinTech have been proposed [9–13].

The design and implementation of a trusted framework which provides the ability to the infrastructure providers to trade their computational and networking resources and to the service providers to negotiate, in real time, and purchase/access the resources with certain SLAs in a trusted environment is extremely valuable. The goal is to enrich the next generation networks with (re-) programmability and configurability, agile resource management optimizing network resource utilization while safeguarding user quality of service. The rapid growth of blockchain and the characteristics of this technology motivated researchers to examine useful use case scenarios in

the area of Next Generation Networks (NGNs). More specifically, blockchain applications in 5G have attracted the interest of academia and industry, as many ideas have been published, [14–16]. Some of those ideas use the blockchain as an immutable database for storing crucial data (i.e., billing, roaming charges), while some others use this technology to guarantee that certain SLAs among systems' entities are met using Smart Contracts (SCs).

This paper aims at: (a) investigating the suitability of blockchain/DLT technologies for flexible and distributed resource management in NGNs, (b) provide a definition of such a blockchain-enabled solution (c) the description of our evaluation test-bed, and (d) the presentation of the initial evaluation results which prove its feasibility and current limitations. To serve this aim, this paper is organized as follows: in Section 2 we survey existing (centralized) approaches, targeting flexible resource management in NGNs, as well as existing distributed blockchain-based approaches that have been proposed up to now. Then, in Section 3, we define the system architecture under consideration, and we present a blockchain-enabled solution based on the Ethereum-Quorum [17] platform, targeting inter-administrative island operation. In Section 4, we present the test bed we deployed to evaluate the approach and present the first results. Finally, Section 5 concludes the paper.

## 2. Relevant Work

Both for inter- and intra-administrative domain resource negotiation and allocation, two main options exist: centralized and decentralized. Typical centralized approaches have been studied and used in many technologies while decentralized solutions are becoming extremely popular in the technological arena. Centralized approaches [18–20] have the advantages that they can achieve high performance due to the availability of information regarding the status of the whole network/domain. However, these come with certain drawbacks: The centralized nature of the brokering mechanism automatically labels it as a SPoF (Single Point of Failure). If this centralized entity is out of service, then the operation of the whole system is disrupted. Furthermore, the communication between the entities participating in the resource brokering should be secured so that the data cannot be altered (which would cause service unavailability).

On the other hand, by adopting the blockchain concept, the different resource providers would be members of the DLT network hosting one (or more) nodes which obviates the SPoF attack. Each of these nodes keeps a copy of the ledger and is participating in the consensus procedure for validating the information registered in the form of transactions. The consensus mechanism used in blockchain discourages any node from performing malicious actions and validating false transactions. Table 1 summarizes the key differences in matters of security and scalability, between centralized and DLT-based resource allocation approaches.

**Table 1.** Centralized vs. decentralized resource allocation.

| Approaches | Security | Scalability |
|:---:|:---:|:---:|
| Centralized Brokering | Low | High |
| Blockchain-based Brokering | High | Low |

*DLT-Enabled Resource Management in NGN*

Blockchain has achieved great popularity since its first application, Bitcoin [21], which introduced the term of digital currency, back in 2009. Following this, today blockchain applications proliferate and address diverse use cases, managing to cover many sections of the modern life apart from digital economy. Solutions in supply chain management [22–24], healthcare [25–27] and education [28–30] are only some of the areas where blockchain is expected to highly contribute and boost performance. Blockchain has managed to showcase its many advantages, especially when combined with Internet of Things (IoT) devices that can initiate transactions that are safely guarded in the blockchain, or crowdsourcing applications [31,32]. However, it has disadvantages too, which are mainly focused

on the scalability limitations that are in place, considering the large amount of data that could be transferred (or stored) in the network.

To address these possible scalability limitations, DAGs have been extensively studied lately, with IOTA [33] being a very popular solution that enhances the performance and scalability of the network when dealing with data received from IoT devices. This performance boost is achieved by not forcing each node to hold the whole ledger—only the transactions that include it and the ones that they have been confirmed by it—limiting the storing requirements of each node while resulting in increased transactions per second (TPS) achieved, [34] and increased support of a larger number of nodes.

Moreover, following their initial use, the majority of the solutions and the platforms (Bitcoin, Ethereum [35], IOTA) included the use of a crypto currency (or token) that facilitated the transactions in the network. On the other hand, platforms like Hyperledger Fabric [36] that do not include a native crypto currency are also rising, especially for commercial use where trusted parties are involved, creating a private blockchain-based network. It is believed that a universal solution does not exist but rather depends on the use case at hand.

Vukolic et al. in [37], present a model for collaborative blockchain-based video delivery. This work studies how the combination of a smart contract stored in a blockchain and network service chaining can be used for supporting collaboration schemes. A keystone of this study is the introduction of a decentralized brokering mechanism for the creation of content sessions through the collaboration of CP (Content Provider) and a TE (Technical Enabler). Then, an attempt for using dynamic service chains takes place in order to benefit from link diversity of different TEs. The decentralized brokering mechanism is established among a CP and a TE which compete and collaborate for the instantiation of the best content delivery session. This decentralized mechanism is based on the blockchain technology and the various stages of this model are described by the use of Smart Contracts (SCs). One of the most critical aspect of the proposed solution is the time needed to converge toward the optimal Content Delivery Contract (CDC), involving the end user, the CP, and the TE. Therefore, authors chose the Hyperledger Fabric blockchain solution, that uses the practical Byzantine Fault Tolerance (pBFT) consensus algorithm, due to its high performance in terms of throughput and latency [38]. Additionally, Hyperledger Fabric (HLF) is a permissioned platform (i.e., every node is known to the other), which is something useful for this particular use case. Although authors present encouraging results in terms of convergence, the scalability of their proposed model based on HLF is questioned, as the experiment nodes were located in the same availability zone of the cloud infrastructure. However, the proposed solution does not specify where these blockchain nodes are hosted nor how they use their wallets for performing transactions in the blockchain network. Furthermore, after ending up with the optimal CDC it is not clear if the SCs in the blockchain are responsible for placement of the required network service function chain for supporting the content delivery.

Rebello et al. in [39], propose a blockchain based solution for secure orchestration operations in virtualized networks, ensuring auditability, non-repudiation and integrity. BSec-NFV Orchestrator (BSec-NFV) aims to protect the creation, management and termination of virtual machines, virtual network functions, and service chains. The contribution of this study lies in the introduction of blockchain and transaction models that provide traceability in a multi-tenant and multi-domain NFV environment. Their use case scenario is based on four key assumptions: (i) limited number of identified providers, as each provider takes part in service level agreements with tenants and other providers; (ii) low number of crash failures, due to the high availability of big data centers; (iii) high throughput and low latency in end-to-end communication, as VNFs are implemented in the network core; and (iv) tolerance to malicious behavior between competing providers and tenants. The authors develop their solution using the HLF that utilizes the pBFT consensus. Their evaluation shows that the overhead added by blockchain is not significant (causes an additional 3% delay with a confidence interval of 95%) while the throughput is considered by the authors to remain in acceptable levels. However, the evaluation is conducted in a data center environment where the various blockchain modules are placed in nearby virtual machines. In NGNs the number of providers may increase and therefore the

tolerance to malicious participants should be higher. Additionally, this work does not focus on the resource negotiation among providers and how this can be achieved using blockchain.

Nour et al. in [40] propose the use of DLT in Network Slicing by presenting a blockchain-enabled Network Slice Broker (NSB). The purpose of the NSB is to guarantee the construction of secure end-to-end network slices in order to support applications of 5G vertical industries, using resources from different stakeholders of the 5G network. When a slice provider receives a request to build an end-to-end slice, it publishes in the blockchain a request for resources regarding each sub-slice composing the end-to-end slice. After receiving the different offers for each sub-slice, the slice provider selects the best offer in terms of cost and the capabilities to meet the requested performance. The proposed solutions introduce the use of two blockchains, one permission-less and one permissioned. The negotiation regarding the resources takes place on the permission-less blockchain, where the prices and capabilities of all offers are visible to everyone. Once the selection of the provider has been made, the permissioned blockchain is used for the creation of the end to end service chain. This work examines the use of Hashcash blockchain which utilizes a Proof of Work (PoW) consensus, and the results present its poor performance in terms of time needed to instantiate a slice. Additionally, authors do not mention which platform they recommend for the public blockchain and which for the private. Moreover, the use of wallets is not examined, and the experiments take place in machines located in the same area which automatically excludes network related parameters in performance evaluation.

Rebello et al. in [41] propose a blockchain solution for network slicing, where they introduce the use of different blockchains for different slice requirements. So, in this work, the network slices are categorized based on their requirements and, the blockchain data structure, the consensus, and the communication protocol are tailored to each specific network slice functionality. The goal of this work is to present a blockchain architecture for the creation of secure network slices for each end-to-end use case in 5G. The implementation of this solution is based also in the HLF software. Similar to previews studies, here the authors propose their solution in data center environments where there is no restriction regarding the resources. To ensure justice in consensus, each data center of the NPs may host at most one blockchain node per blockchain (it is reminded that each slice type is associated with a different blockchain). Blockchain nodes in a slice type are invisible to anyone outside the slice. This study proposes the use of a management blockchain where all VNF orchestration operations are logged in order to provide auditability and management regarding the slice creation. The management of various VNFs is accomplished by using SCs to introduce transparency and automation in this decentralized system. The architecture of this system is composed by four components: a user interface, the NFV MANO module, a blockchain creation server module, and a management blockchain server module. However, the evaluation of the prototype is conducted in one physical machine where the HLF nodes are running inside a container. As a result, we cannot be sure how this solution would operate if the nodes were in different networks and locations. Additionally, this work assumes that the blockchain runs in a data center environment. Furthermore, the scalability of the presented solution is not well defined, although in contrast to previews works, here a detailed analysis of blockchain's operation is illustrated. Finally, authors are not focusing on the resource negotiation procedure that takes place among providers in this multi-tenant and multi-domain NFV environment. Finally, Table 2 summarizes this section of the paper by illustrating the key characteristics of the examined proposals while it presents the profile of the ideal solution.

**Table 2.** Current studies and ideal solution.

| DLT and NGN | Blockchain/ DAG | Resource negotiation/ management | Network Slicing | Cloud environment/ Data Center | VNF management/ service chains | Consensus | Permissioned / Permissionless | Use of multiple blockchains |
|---|---|---|---|---|---|---|---|---|
| Vukolic et al. [37] | Blockchain | Y | - | Y | Y | PBFT | Permissioned (HLF) | - |
| Rebello et al. [39] | Blockchain | Y | - | Y | Y | PBFT | Permissioned (HLF) | - |
| Nour et al. [40] | Blockchain | - | Y | Y | Y | PoW | Permissionless (Hashcash) | - |
| Rebello et al. [41] | Blockchain | - | Y | Y | Y | PBFT | Permissioned (HLF) and Permissionless (unspecified) | Y |
| Proposed Solution | Blockchain | Y | Y | Y | Y | RAFT | Permissioned | - |

The afore mentioned works confirm the interest of academia regarding the combination of NGN adopting SDN/NFV for flexible resource allocation and Blockchain. Even though, this research domain is immature, the benefits that 5G industry may acquire by the development of this kind of applications are tremendous. This type of solutions offers to NPs an environment of trust without the need of a trusted third party, where the security of transactions is guaranteed, and all actions are accountable. As a result, companies and organizations will feel safer to form a federated environment, due to blockchains' key attributes.

However, there are certain challenges that need to be addressed to persuade the actors of the sector about its suitability: resource consumption, transaction per second (TPS) rate, overhead, scalability. Additionally, the time needed for performing the correct adjustments in the network is critical in NGNs, therefore this is an important variable of this solution. In contrast to related works, the current study provides a proof of concept of applying blockchain for resource management (including the various categories described in Table 2) accompanied with evaluation results. The novelty of this paper lies in the evaluation of a real blockchain network where a SC that describes a resource management scenario is used. The results show that the use of blockchain for resource management is feasible.

## 3. System Architecture and Use Case Scenario

Infrastructure operators are becoming totally separated from service providers, while the life cycle of each network service is becoming shorter and services become more and more demanding in terms of network dynamicity, computational capabilities, and flexibility [42]. In order to offer high quality services under highly varying load patterns due to high mobility and data-intensive tasks, the deployment of services over network infrastructures should be decided as dynamically and flexibly as possible and, more importantly, across the boundaries of networks belonging to different administrative areas.

The system architecture we are proposing is shown in Figure 1. We assume that several NPs exist, each operating a MANO instance to orchestrate the use of its own resources which consist of one or more NFV Infrastructures (NFVIs). In each one of the MANO instances, the corresponding monitoring component [43–45] is aware of the level of resource utilization, as well as of the quality of service experienced by the deployed services. Currently, this component can trigger the re-configuration of the resources that the specific MANO administers, including "leased' resources which are statically defined upon agreement. For the different administrative areas to "trade" resources in real time, either a central trusted authority or a distributed solution should be in place. In Figure 1, a solution where each area (using a MANO instance) is maintaining a blockchain node (BC i) is shown. Each BCi shown in the figure is assumed to host the wallet of the blockchain solution and the blockchain node that contains the digital ledger. This entity is triggered by the MANO when a need for additional

resources or the availability of resources is detected by the monitoring component. This way resource pooling across administrative areas becomes possible without the need for any trusted 3rd party.

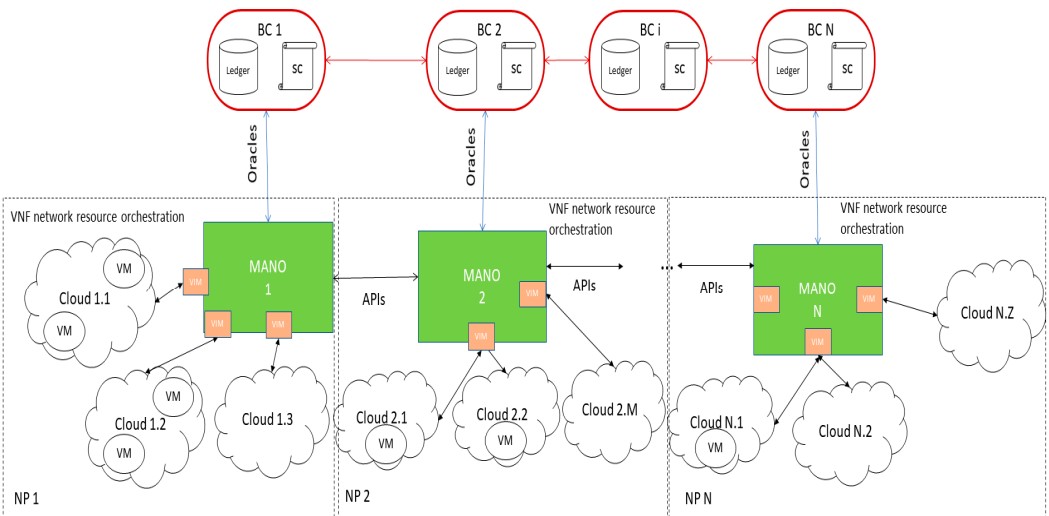

**Figure 1.** The proposed system architecture.

The role of each component depicted in Figure 1 is presented in detail in the following bullets:

- *Blockchain nodes*: The blockchain nodes form the blockchain network and hold the digital ledger that contains all the history of transactions and information regarding the NPs. Each node has, also, access to the wallet of the NP that contains the keys of the node which are needed to access the Blockchain network and make the required calls to the SCs deployed there. The blockchain network is responsible for the proper functionality of the resource management mechanism, that includes the trade of resources and their billing. Each NP supports one node as it is presented in Figure 1.
- *Oracles:* A blockchain oracle [46,47] is an entity that connects a blockchain with off-chain data. Oracles are known as blockchain middleware and enter every data input through an external transaction. To maintain the deterministic validation of blocks, normally smart contracts can only access data previously stored on the blockchain and cannot use external data. The use of oracles makes communication possible from the external world to the blockchain, for example by recording external data on the blockchain in transactions. In the presented solution, oracles are used for the interaction of the Blockchain network with the MANO components and the VNF network resource orchestration.
- *MANOs:* These components are responsible for performing the necessary actions for the implementation of the resource management. The resource management mechanism is implemented inside the blockchain using the SC and the MANO components execute the decisions derived by the blockchain network. Additionally, this component is responsible for monitoring the resource utilization of the virtual infrastructure and hosting images of virtual network functions (VNFs). The MANOs interact each other to reserve resources and implement the necessary network functions specified by the blockchain network which acts as a decentralized brokering system.
- *VIMs:* Virtual Infrastructure Manager (VIM) is responsible for managing the virtual infrastructures, usually cloud environments, and is hosted inside the MANO component. Through VIM, MANOs can manage these resources by launching, modifying, and terminating VMs that support various VNFs.
- *Clouds:* Cloud infrastructures offer computational resources to support various VNFs. These infrastructures are geographically staggered in order to cover regions and cities, trying to provide services near to customer.

- *VMs:* The VMs are used for hosting the VNFs and are consisted of virtual resources such as VCPUs, RAM, storage, and network links (bandwidth). The resources used by the VM are based on the characteristics of the VNF.

The proposed solution introduces the benefits of Blockchain (BC) technology in a federated environment consisting of NPs. The basis of the solution is a Smart Contract (SC) written in solidity and deployed in a Quorum network. Quorum is a fork of Ethereum and was selected because it can support private transactions using the Tessera tool [48] and can be easily implemented using different consensus mechanisms. The SC consists of three main functions:

(a) addNetworkProvider: This function is triggered each time a new NP joins the network. For every NP, the information kept in the BC includes: the name of the NP, the types and number of its offered resources, e.g., bandwidth, processing, memory, the cost of the resources per unit, other attributes of the resources like the region they cover and (most importantly) the Service Level Agreement (SLA) that the NP can support. A unique blockchain account address is also associated with each NP and is used as a wallet for interacting with other NPs and entities of the blockchain. This function performs a transaction and inserts the result in the ledger which is kept in the blockchain.

(b) GetBestMatch: When an NP needs additional resources to satisfy the needs of its users, it searches the BC network in order to find another NP that can offer these resources. The GetBestMatch function is triggered in this case. It takes as an input the type, number and attributes of the needed resources and searches to find the NP that can fulfil them. In case more than one NPs can satisfy the request, the one incurring the lower cost is selected. It is worth stressing here that (a) our focus is not on the optimization algorithm but on the evaluation of the feasibility of such a solution offering adequate performance, and (b) the "cost" that we assume in the proposed solution can be the actual financial cost or any other metric, whose value is designed to be minimized. It should be noted that this function reads data from the blockchain and does not write any information in the ledger.

(c) ResourceReservationTransaction: Once the GetBestMatch function ends up with the id of the NP that offers the required resources at the lowest price, the ResourceReservationTransaction is triggered so that the decision and relevant payment are enacted. The NP that has requested resources pays the amount specified by the cost field of the NP who offers the resources and the resources of the provider that purchased the resources increase. The balance of the NP that has offered resources increases and the transaction is completed. This transaction function writes data into the blockchain.

In a commercial solution, an additional function that will trigger the release of the resources would be implemented. It should be noted that, the use of cryptocurrency for the billing is not examined in this solution but is a mechanism that could be included as a feature in future extensions. The billing process of the solution uses as input the utilization of resources and the features of those resources described by the SLA.

In order to illustrate the functionality of the proposed solution, a use case scenario is described as follows. Each NP registers into this solution by using the addNetworkProvider function of the SC and becomes member of the blockchain by maintaining a node. Each NP uses its own resources to support its own customers and the resources that are not in use are available to the network. When a NP needs resources to cover an increased demand, creates a call at the GetBestMatch function of our Smart Contract to select the proper NP among candidates. The proper NP is the one that offers the required resources in the lowest price. The outcome of the GetBestMatch function is used by the third function (ResourceReservationTransaction) in order to initiate a transaction between the NP that has requested resources and the one that offers them. When the transaction function is triggered, the NP that has requested resources (NPreq) transfers an amount of digital money, which are called ethers for our Quorum implementation, to the NP that provides (NPprov) resources. The NPprov lends resources

to the NPreq and immediately uses an Oracle mechanism in Blockchain terms to implement the necessary services using the MANO components. MANO components are responsible for managing (e.g., launching, terminating) the required network services using the resources specified by the SC. The computational resources are offered by the cloud infrastructures through the VIMs (Virtual Infrastructure Managers) and are used for the creation of VMs which support the network services described by MANO. When the NPreq does no longer need the borrowed resources, these resources are released back to the original owner while the responsible MANO terminates the reserved services.

Having in mind that next generation networks use microservices and the lifetime of those services vary (from ms to seconds or even minutes) the above process should be performed with the minimum latency while the number of supported transactions should be high. Therefore, the next section evaluates this solution and checks its feasibility and its performance in matters of latency and throughput. It is worth mentioning that the current work illustrates a proof of concept of a blockchain application for dynamic resource management in next generation networks. The technical details for the interaction of blockchain with the virtual resources, are out of the scope of this paper and will be studied further in the future. The use of APIs and blockchain oracles for the interaction of blockchain with other technologies are very promising and will be tested in future version.

## 4. Evaluation Results

To evaluate the proposed approach, we have deployed it a custom Quorum network. The goal of this section is to evaluate the feasibility of this solution and its performance in terms of transaction latency and transaction throughput, as well as the number of transactions that can be handled by the network to check the load burden. The test bed that we set up included three nodes which are not hosted on the same physical machine (as is done in all the surveyed articles discussed in this paper) but in machines interconnected through the Internet. We also deployed a fourth VM used for running Hyperledger Caliper, which is our benchmarking tool [49]. The Hyperledger Caliper tool is one of the most popular benchmarking solutions for blockchain applications. Caliper uses an adapter to connect to the System Under Testing (SUT) which, in this case, is the private Quorum network based on the RAFT consensus. Figure 2 illustrates the basic components used in the presented experiment. The Load Generator produces the load applied to the SUT, while the Configuration file describes the experiment. The Adapter is used for the interaction with the SUT which in our case is the Quorum Network. The outcome of the whole experiment is the report file which contains information related to the behavior of the network. The Quorum Nodes that form the network are hosted in the Okeanos cloud [50] infrastructure offered by GRNET. The characteristics of the VMs are:

- Operating system: Ubuntu 16.04 LTS server,
- 4 CPU cores,
- 8 GB RAM,
- 30 GB storage, and
- public IP addresses.

The performance evaluation was conducted as a function of three different parameters which were configured through a YAML file. These parameters included: (a) the number of workers, (b) the rate controllers, and (c) transaction number (txNumber). The workers are docker containers which generate the workload in the network. The rate controllers are two parameters affecting the rate at which load is inserted in the blockchain network. They take under consideration TPS which is the number of transactions to be sent in a second and txDuration which specifies the duration till which we will be sending the transaction. The txNumber is the number of transactions to be executed and represents the amount of transactions initiated when the functions of the SC are executed. In our experiments, we used one worker and the txNumber was set at very high values to ensure that we measure the steady state at different input loads (different TPS values).

The output metrics of the test bed which we measure are three: (a) Success or Fail of a transaction, (b) the average Transaction Latency (sec) and (c) the average Transaction Throughput (TPS). Every transaction that has successfully been processed, verified and inserted into a block is considered as a Success. The Transaction Latency is defined as the time elapsed between the initiation of a transaction (submit transaction) to the time the transaction has been completed and *inserted into the blockchain.* When the transaction has been inserted to the blockchain it is available to the whole network (all nodes). In other words, *Transaction Latency = Confirmation time − Submission time.* Finally, Transaction Throughput is the rate at which valid transactions are committed by the blockchain SUT not in terms of a single node, but across the entire system. It is stressed that this is expressed as transactions per second but should be distinguished from the TPS which is an input parameter for each scenario we execute and is the rate of transaction submission to the system. In other words, *Transaction Throughput = Total committed transactions/total time in seconds.*

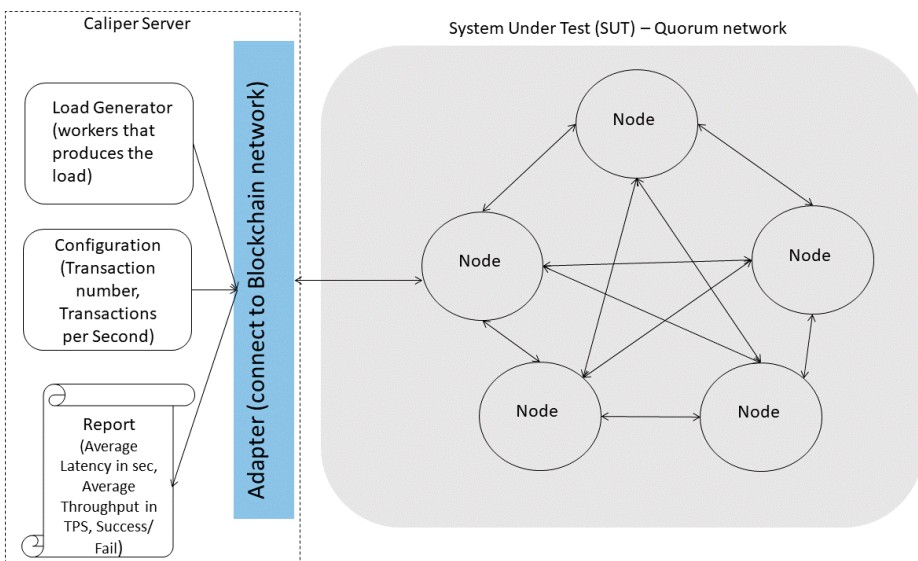

**Figure 2.** The experimental testbed.

We have run a set of scenarios changing the number of (input) transaction per second parameter per smart contract function to assess (a) how many transactions the solution can handle, (b) the time required for a resource transaction to be decided and stored in the blockchain and (c) on the processing and memory resources required for the implementation of the solution.

For the first metric of interest, throughput (i.e., transactions stored in the blockchain per second), the results are shown in Figure 3. As the number of transactions (submitted to the system) per second increases the throughput (i.e., transactions that were successful and stored in the blockchain) increases as well. This is true up to 10TPS while from this point on, the number of transactions stored in the blockchain (reflected in the vertical axe) do not increase any further. The fail is attributed to the continuously increased latency of each transaction to be successfully executed. The response time of the transaction exceeds the Caliper's acceptable time limit and, therefore, is characterized as failed. As a result, the number of failed transaction increases inevitably as the (input) transaction rate becomes higher.

This is also proven by the results for latency, which are shown in Figure 4. The latency (in seconds) of the two functions, that do not require any reading or processing (addNetwork provider and Reservation transaction mentioned in the figure as transaction), is kept very low irrespective of the TPS. This is not the case for the getBestMatch which requires significant processing. For the getBestMatch function, the latency is very low as soon as the TPS is below 4 and increases to 15 sec when TPS becomes 10.

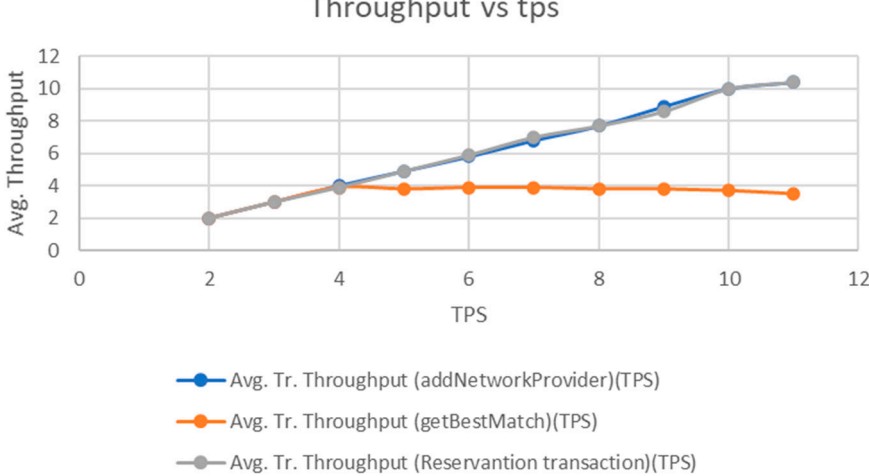

**Figure 3.** The throughput per smart contract function.

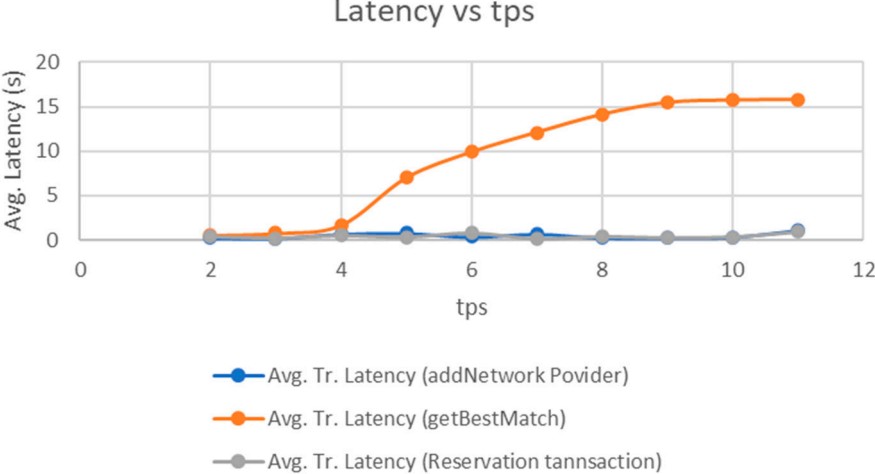

**Figure 4.** The latency per smart contract function for different transaction loads (submitted transactions per second).

This is a very important result. It means that if a group of NPs decide to allow for up to 4 resource reallocations per second, the re-configuration of the resource allocation will be decided in less than 1s which is a very low latency result and leads to agile network reconfiguration. When the number of TPS increases to 10 the situation becomes like the one expected with more NPs joining the system or having fewer providers but with more than one resource requests per second. In this case, the latency becomes 15s which can still be considered acceptable assuming these requests correspond to pipes of traffic among service providers and not as single end-user services. To consider flows of finer granularity, we need to improve the solution. From the value of 10TPS and above, fails in the transaction occur (5% at TPS equal to 10 and rising with TPS). So, the presented solution can offer adequate performance up to 10 TPS. In both latency and throughput results, we presented the average values i.e., the average over the multiple runs we executed.

With respect to the resources needed for the implementation of the solution, in the aforementioned test bed, a max CPU utilization of 35% and of 1.5 GB memory is measured which is definitely affordable. Finally, the results produced in this section show that the impact of the SC on the Quorum blockchain network is not significant in terms of latency and throughput. The benefits provided by introducing the blockchain technology and the results of the presented evaluation show that the use of blockchain technology for resource management is feasible and promising.

## 5. Conclusions

Summarizing the work presented in this paper, the idea of adopting distributed, blockchain-enabled solutions in modern networks has triggered the interest of research community. The application of blockchain technology adds valuable characteristics to modern networks as it forms a network of trust among participants, while it guarantees the integrity of the information stored and used by the system.

In the presented solution, we proposed the use of a distributed broker mechanism by describing an architecture that includes the NPs as nodes, oracles for interaction within and out of the blockchain network and wallets to send and receive transactions. MANOs, VIMs, VMs and cloud instances provide for a complete view of the overall architecture that aims to showcase the strength of distributed solutions following a described use-case scenario that underlies its potential. Following the described architecture and the use case in hand, basic blockchain characteristics such as transparency, immutability, non-repudiation are examined as to whether they can provide for a safe multitenant environment for the NPs to perform resource management processes without relying on a trusted, centralized third party. Additionally, this concept opens the road for the formation of new business models between NPs which can reduce their cost and at the same time optimize management of their resources. The proposed solution requires from each of the NPs to host a blockchain node, to support the presented logic. In contrast to other related works, this paper describes and evaluates the blockchain based resource management solution and produces results regarding its feasibility and performance by applying Caliper, a well-known Blockchain emulator, to perform this task. The results show that the cost of the solution is more than affordable on one hand, while on the other the achieved performance, even when the solutions is not optimized, is adequate. According to the results, the latency of resource reconfiguration decisions remains below 15s for high loads, and the throughput is also adequate.

In our future work, we are planning to examine the deployment of a larger blockchain network, consisted of more nodes to test the scalability of our solution. We will, also, implement more sophisticated resource allocation algorithms to evaluate their impact on the latency and we will work on the SC and the role of the oracles, described in our architecture. Additionally, we will study the impact of the adopted consensus algorithms on the achieved results by applying different consensus mechanisms and test the performance changes on each one of them in an effort to identify an optimum implementation for our solution. In addition to evaluate the effect of using a distributed brokering mechanism and to measure the impact of blockchain (the traffic it generates) in the overall network performance, solutions that are based on DLT will also be examined.

**Author Contributions:** M.X. has contributed in the investigation of the state of the art, contribution in the design of the solution, development of the solution in software and of the test-bed. D.G.K. contribution focused on the design of the software architecture and the validation of the solution. P.K. contributed in the use case definition, in the definition of the smart contracts functionalities and in the analysis of the MANO needs and benefits from the solution. H.C.L. established the methodology to carry out the work and contributed in the solution design. C.P. supervised of the overall work and contributed with review and editing of the paper. All authors contributed in the writing of the paper. All authors have read and agreed to the published version of the manuscript.

**Funding:** All authors would like to thank the University of West Attica for the financial support provided to them to undertake this research project.

**Acknowledgments:** All authors would like to thank the University of West Attica for the financial support provided to them to undertake this research project.

**Conflicts of Interest:** The authors declare no conflict of interest.

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
