# Peer review of "Application of Blockchain Technology in Dynamic Resource Management of Next Generation Networks"

_information, doi:10.3390/info11120570_

Round 1

Reviewer 1 Report

This paper proposes a blockchain-based solution to handle the negotiations with network providers. In particular, given the the requirements of a virtual infrastructure, the solution will take care of identifying the providers that offer compatible services at the best price. Also, once a specific provider is selected, the solution will take care of reserving the resources required for the virtual infrastructure and handling the payments.

The idea behind this paper is definitely interesting, and the analysis of the state of the art has been performed exhaustively. However, the novelty of this contribution with respect to the related work is unclear. In particular, the authors claim that their solution is capable of managing various aspects of virtual infrastructures, such as VNFs, network slicing, VMs etc. Yet, in the description of their solution, they fail to provide any information on how they handle such resources. Indeed, only a very high-level diagram of the architecture of their solution is present, whereas any detailed information, smart contract, source- or pseudo-code snippet is absent.

Also, information on how resources are billed to the customer is rather scarce. Does the solution rely on cryptocurrency for the payments? Do customers have to pay in advance, or only per-use? What happens if a customer runs out of money?

Finally, the advantages brought by the blockchain in this specific solution should be better clarified. Do the authors exploit blockchain for the immutability and transparency it provides? Do they use it for the possibility to pay with cryptocurrency?

To sum up, for this contribution to be significant, the authors must provide more detailed information on how their solution works. To this aim, a real-world use case showcasing the solutions' features would help.

Author Response

The comments and suggestions of the reviewer were extremely valuable and led us to improvements that leveraged the quality of the manuscript. In the attached file a point-by-point response is presented.

Reviewer 2 Report

The paper presents the use of blockchain technology for dynamic management of resource in next generation networks.

The weak points of the paper are as follows:

  1. almost half of the paper contain the state-of-the-art
  2. the proposed system and its architecture are described very poor - given very general, small and not visible shcema without any description and 3 functions with a small description is really too little
  3. name of section 3 is "System architecture and use case scenario" but I didn't find any use case scenario inside this section
  4. section 4 is "Evaluation results" but it is not clear what was evaluated
  5. figure 2 is not clear
  6. figures 3 and 4 are too small
  7. conslusion is poor
  8. references have to be improved

Author Response

(The authors gave the same response as above.)

Round 2

Reviewer 1 Report

The authors have addressed all my concerns. I have no further comments on this work.

Author Response

We would like to thank the reviewer for his/her valuable suggestions. The comments and suggestions of the reviewer leveraged the quality of the manuscript.

Reviewer 2 Report

Some of my previous comments are not realized. Still:

  1. almost half of the paper contains the state-of-the-art (about 6 pages for 14)
  2. the proposed system and its architecture are poor described
  3. conclusions are poor
  4. references have to be improved

Author Response

We would like to thank the reviewer for his/her valuable suggestions. The comments and suggestions of the reviewer leveraged the quality of the manuscript. The revised document presents the changes we have made in order to answer to all suggestions and comments.

Round 3

Reviewer 2 Report

My previous comments - requirements to improve the paper were realized.